# Less absorbed solar energy and more internal heat for Jupiter

Liming Li[1], X. Jiang[2], R.A. West [3], P.J. Gierasch[4], S. Perez-Hoyos[5], A. Sanchez-Lavega[5], L.N. Fletcher[6], J.J. Fortney[7], B. Knowles[8], C.C. Porco[8], K.H. Baines[3], P.M. Fry [9], A. Mallama[10], R.K. Achterberg[11], A.A. Simon[12], C.A. Nixon[12], G.S. Orton[3], U.A. Dyudina[13], S.P. Ewald[14] & R.W. Schmude Jr.[15]

The radiant energy budget and internal heat are fundamental properties of giant planets, but precise determination of these properties remains a challenge. Here, we report measurements of Jupiter's radiant energy budget and internal heat based on Cassini multi-instrument observations. Our findings reveal that Jupiter's Bond albedo and internal heat, $0.503 \pm 0.012$ and $7.485 \pm 0.160$ W m$^{-2}$ respectively, are significantly larger than $0.343 \pm 0.032$ and $5.444 \pm 0.425$ Wm$^{-2}$, the previous best estimates. The new results help constrain and improve the current evolutionary theories and models for Jupiter. Furthermore, the significant wavelength dependency of Jupiter's albedo implies that the radiant energy budgets and internal heat of the other giant planets in our solar system should be re-examined. Finally, the data sets of Jupiter's characteristics of reflective solar spectral irradiance provide an observational basis for the models of giant exoplanets.

[1] Department of Physics, University of Houston, Houston, TX, 77004, USA. [2] Department of Earth and Atmospheric Sciences, University of Houston, Houston, TX, 77004, USA. [3] Jet Propulsion Laboratory, California Institute of Technology, Pasadena, CA, 91109, USA. [4] Department of Astronomy, Cornell University, Ithaca, NY, 14850, USA. [5] Departamento de Fisica Aplicada I, Escuela de Ingenieria UPV/EHU, Bilbao 18013, Spain. [6] Department of Physics and Astronomy, University of Leicester, Leicester LE1 7RD, UK. [7] Department of Astronomy and Astrophysics, University of California, Santa Cruz, CA, 95064, USA. [8] CICLOPS/Space Science Institute, Boulder, Colorado 80301, USA. [9] Space Science and Engineering Center, University of Wisconsin-Madison, Madison, WI, 53706, USA. [10] Department of Mathematics and Statistics, University of Maryland, College Park, MD, 20742, USA. [11] Department of Astronomy, University of Maryland, College Park, MD, 20742, USA. [12] NASA Goddard Space Flight Center, Greenbelt, MD, 20771, USA. [13] Space Science Institute, Boulder, CO, 80301, USA. [14] Division of Geological and Planetary Sciences, Caltech, Pasadena, CA, 91125, USA. [15] Gordon State College, Barnesville, GA, 30204, USA. Correspondence and requests for materials should be addressed to L.L. (email: lli7@central.uh.edu)

The radiant energy budgets of planets and satellites, which are mainly determined by the absorbed solar energy and the emitted thermal energy, play critical roles in the thermal properties and evolution of these astronomical bodies[1,2]. For bodies with significant atmospheres, the radiant energy budgets at the top of their atmospheres set critical constraints for the total energies of the atmospheric systems. Furthermore, the transfer and distribution of radiant energy within the atmospheric systems modify the thermal structure and hence generate the mechanical energy to drive atmospheric circulation, weather, and climate[3,4]. The radiant energy budget and the related internal heat of the giant planets also bear upon their evolutionary history[5–12].

The radiant energy budgets of planets and satellites in our solar system have been explored for a long time. For some terrestrial bodies (e.g., Earth and Titan), the absorbed solar energy basically balances the emitted thermal energy at a global scale[13–15], even though small energy imbalances probably exist and contribute to the climate change on these terrestrial bodies[16–20]. Significant imbalances between the emitted thermal energy and the absorbed solar energy have already been discovered for some giant planets[1,2,21–23]. Therefore, an extra energy source, which is referred to the internal heat or intrinsic flux[6,7,10–12], is inferred. The internal heat comes from the secular cooling of the planet's interior from an initially hotter state postformation[6], with a possible additional component due to release of gravitational energy due to the separation of chemical components[6,8–12]. These processes provide important clues for understanding planetary formation and evolution.

However, the previous studies of the radiant energy budgets of giant planets are based on observations with limited coverage of viewing angle and wavelength. The observations of Jupiter and Saturn conducted by the Cassini spacecraft have many advantages over previous observations. Therefore, we can measure Jupiter's radiant energy budget and internal heat with unprecedented precision.

## Results

### Jupiter's albedo spectra in the domain of phase angle and wavelength.
In this study, the multi-instrument observations obtained by the Cassini spacecraft during its Jupiter-flyby mission (2000–2001) are used to investigate the radiant energy budget and internal heat on Jupiter. The methodology of energy budget measurements has been well described in previous studies[1,2,24–26]. The infrared observations conducted by the composite infrared spectrometer (CIRS)[27] and the visual and infrared mapping spectrometer (VIMS)[28] on Cassini have already been used to measure the emitted power of Jupiter[26]. To determine Jupiter's radiant energy budget and hence internal heat, we need to determine the other energy component—the absorbed solar power. Here, we provide the measurements of Jupiter's absorbed solar power based on observations primarily recorded by the imaging science subsystem (ISS)[29] on Cassini. These data have much better coverage of phase angle (i.e., a critical factor of measuring the absorbed solar power, see Methods section "Theoretical methodology") than previous observations (e.g., Pioneer and Voyager). Examples of the ISS-recorded global images of Jupiter at different phase angles are shown in Fig. 1.

The theoretical framework for computing Jupiter's absorbed solar power is described in the section Methods. Based on the theoretical framework, we need measurements of total incident solar radiance, geometric albedo, and phase integral to compute Jupiter's Bond albedo over a wavelength range of 0–4000 nm, the wavelengths over which the solar spectral irradiance significantly contributes to the total solar power. The combined data sets from the Cassini observations and other observations provide such measurements (see Methods section "Summary of observational data sets" and Supplementary Table 1). The total solar power is mainly based on Earth-based measurements of solar spectral irradiance in 2000–2001 (see Methods section "Data sets of the SSI" and Supplementary Fig. 1) to be consistent with the observational time of the Cassini mission. The phase function and hence phase integral of Jupiter's albedo spectra (i.e., reflectance spectra) mainly come from Cassini ISS and VIMS observations. The Cassini data sets and the corresponding dataprocessing techniques are introduced in Methods (section "The Cassini ISS/ VIMS observations and data processing", Supplementary Figs. 2–4, and Supplementary Table 2).

The Cassini ISS and VIMS full-disk observations are used not only for the investigation of Jupiter's phase function/integral but also for the albedo spectra (see Methods section "Available albedo spectra from other observations" and Supplementary Fig. 5). Therefore, we first validate the Cassini ISS and VIMS observations by comparing with other available observations (see Methods section "Validation of the ISS and VIMS observations" and Supplementary Figs. 6–8). Then we use least-squares fitting[30] to fill the observational gaps in phase angle (see Methods section "Filling observational gaps in phase angle" and Supplementary Figs. 9–14) and wavelength (see Methods section "Filling observational gaps in wavelength and Supplementary Figs. 15, 16) of the ISS and VIMS data.

After filling the observational gaps in phase angle and wavelength for the Cassini ISS and VIMS observations, we have Jupiter's complete full-disk albedo spectra in the two-dimensional

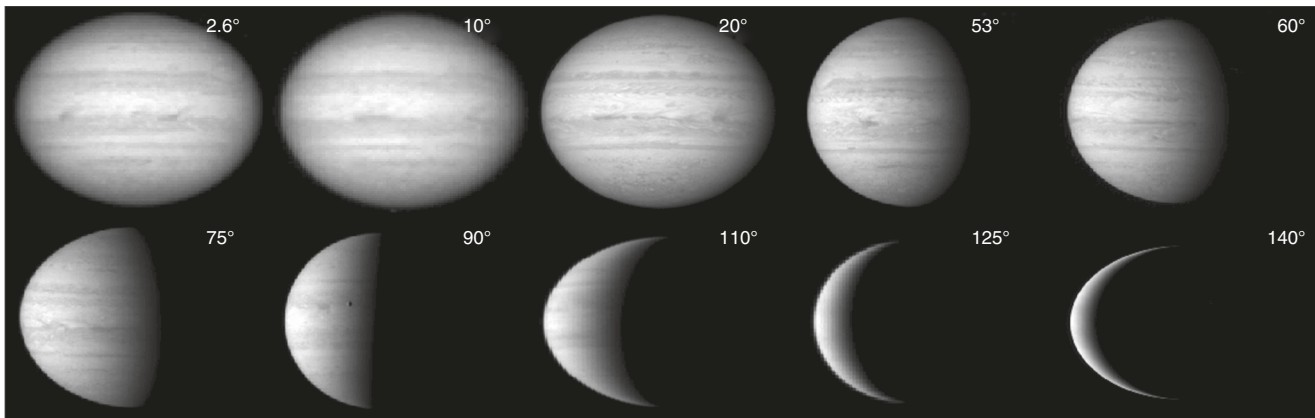

**Fig. 1** Examples of Jupiter's global images in the indicated phase angles recorded by the Cassini ISS. The images were recorded by the Cassini ISS at the second continuum filter (CB2) during the Jupiter-flyby mission in 2000–2001

domain of phase angle and wavelength, as shown in Fig. 2. Based on Fig. 2, we can obtain the monochromatic geometric albedo, the monochromatic phase integral, and the monochromatic Bond albedo (i.e., spherical albedo) at each wavelength, which are shown in Fig. 3. As discussed in Methods (see section "Uncertainty estimate" and Supplementary Fig. 17), the uncertainty from calibrating the Cassini data, the uncertainty in filling the observational gaps, and the standard deviation of multiple measurements are considered for estimating the error bars in Fig. 3.

Panel A of Fig. 3 shows that the monochromatic geometrical albedo reaches a maximum value ~0.652 ± 0.047 at a wavelength ~565 nm. This panel also suggests that Jupiter's monochromatic geometric albedo is relatively large in the visible band from ~200

to ~1600 nm except for a few absorption bands. At relatively longer wavelengths (i.e., >1600 nm), the monochromatic geometric albedo is very small (<0.1) except for wavelengths around 1850 and 2700 nm. Panel B of Fig. 3 suggests that the phase integral varies from ~1.1 to ~1.3 in the wavelengths shorter than ~1050 nm, which is roughly consistent with a previous analysis[31]. At longer wavelengths (>1050 nm), the variation of phase integral becomes larger. The average phase integral is larger at wavelengths longer than 1050 nm (~1.56) than at wavelengths shorter than 1050 nm (~1.25). Figure 3 also shows that the monochromatic Bond albedo can reach a maximum of 0.797 ± 0.069 at a wavelength ~678 nm (panel C). Except for the wavelengths around 1850 nm and 2700 nm, Jupiter's monochromatic Bond albedo is generally larger in the wavelength range of ~200–1600 nm than at wavelengths outside of the range, which is roughly consistent with the wavelength distribution of the monochromatic geometric albedo (panel A).

Based on the distribution of the monochromatic Bond albedo over wavelength (panel C of Fig. 3), we can obtain the wavelength-average Bond albedo (i.e., Jupiter's Bond albedo) by weighting the monochromatic Bond albedo with the solar spectral irradiance in 2000–2001 (Supplementary Fig. 1). The resulting Bond albedo of Jupiter is 0.503 ± 0.012 (see Methods section "Uncertainty estimate"). The best estimate from previous analyses of Jupiter's Bond albedo, which is based on the combined analyses of Pioneer and Voyager[24], suggests a value 0.343 ± 0.032. As discussed in Methods (see sections "The Cassini ISS/VIMS observations and data processing" and "Comparison between the Cassini and previous measurements"), our study is superior to the previous best analysis[24] mainly in three aspects. First, the Cassini camera systems and the radiance calibration are more advanced than corresponding instruments and calibration used in the Pioneer/Voyager spacecraft. Second, the coverage of wavelength and phase angle is more complete in the Cassini observations than in the Pioneer/Voyager observations. Finally, the wavelength-dependent nature of Jupiter's reflected radiance (Fig. 1) is much better addressed in the current study than in

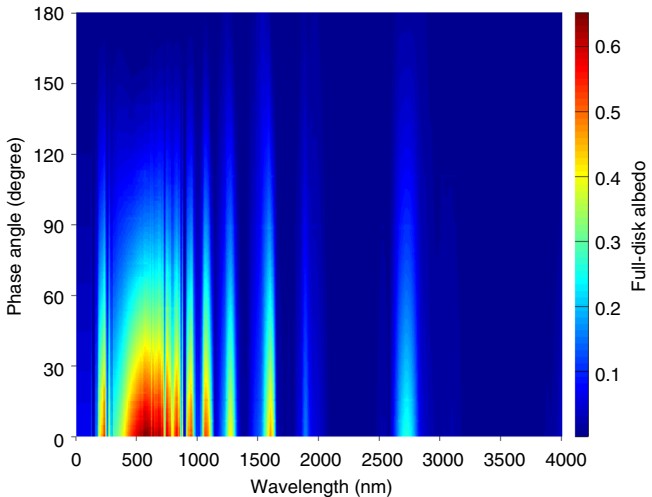

**Fig. 2** Jupiter's albedo in the two-dimensional domain of phase angle and wavelength. The full-disk albedo with varying phase angle (0°–180°) is displayed in the wavelength range of 0–4000 nm, in which more than 99% of the total solar power is concentrated

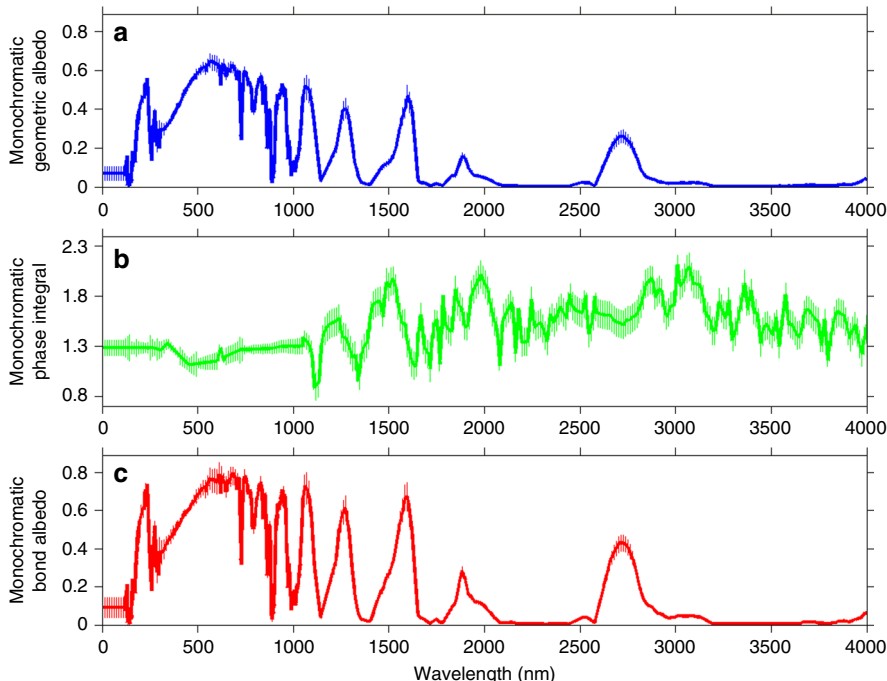

**Fig. 3** Jupiter's monochromatic geometric albedo, phase integral, and Bond albedo. **a** Monochromatic geometric albedo. **b** Monochromatic phase integral. **c** Monochromatic Bond albedo. Vertical lines in the three panels represent error bars of measurements

the previous study. Therefore, the analyses in this study provide the best measurements of Jupiter's Bond albedo. In addition, we rule out the possibility that the significant difference between the Cassini result (0.503 ± 0.012) and the Pioneer/Voyager result[24] (0.343 ± 0.032) is due to possible temporal variation of Jupiter's albedo from the Pioneer/Voyager epoch to the Cassini epoch (see Methods section "Possible temporal variations of Jupiter's energy budget" and Supplementary Figs. 18, 19).

**Jupiter's radiant energy budget and internal heat**. Jupiter's Bond albedo can be used to investigate the reflected/absorbed solar power with the known total solar power (see Methods section "Theoretical methodology"). Integrating the solar spectral irradiance over wavelength and considering the distance between the Sun and Jupiter in 2000–2001, the total solar power at Jupiter (i.e., solar constant) is 53.48 W m$^{-2}$. The solar power is reflected/ absorbed by the projected disk of Jupiter in the Sun's direction. Based on the ratio between the area of the imaginary disk and Jupiter's global area (0.2488, see Methods section "Jupiter's disk area and global surface area"), the global-average solar power is 53.48 × 0.2488 ~ 13.306 Wm$^{-2}$. Therefore, the reflected solar power and the absorbed solar power are 6.693 ± 0.160 and 6.613 ± 0.160 W m$^{-2}$, respectively. Our previous study[26], based on Cassini CIRS and VIMS observations, suggests that Jupiter's emitted power is ~14.098 ± 0.031 W m$^{-2}$ in the Cassini epoch. Therefore, Jupiter's internal heat is estimated to be 7.485 ± 0.163 W m$^{-2}$ by assuming energy equilibrium for Jupiter, which is 37.5 ± 8.8% larger than 5.444 ± 0.425 W m$^{-2}$ from the previous best estimate[24]. Furthermore, the new measurements suggest that the ratio of the emitted thermal power over the absorbed solar power is 2.132 ± 0.051, which is also significantly larger than 1.668 ± 0.085 from the previous estimate[24]. The new results of Jupiter's radiant energy budget and internal heat are summarized in Table 1.

## Discussion

In this study, we have conducted an analysis of Jupiter's radiant energy budget mainly from Cassini multi-instrument observations. The analysis provides a complete determination of Jupiter's albedo spectra, not only by wavelength, but also by phase angle for the first time (Fig. 2). Therefore, we can measure Jupiter's albedo and the related radiant energy budget/internal heat with an unprecedented precision. Our measurements suggest that Jupiter's Bond albedo and internal heat (0.503 ± 0.012 and 7.485 ± 0.160 W m$^{-2}$, respectively) are significantly larger than the previous best results (0.343 ± 0.032 and 5.444 ± 0.425 W m$^{-2}$) based on the Pioneer and Voyager observations[26].

The significant increase in Jupiter's internal heat found from our analyses necessitates a re-examination of three decades of models of the thermal evolution of the planet. Models of Jupiter that have included both its secular cooling and the implied depletion of helium into the deep interior[32,33] have always produced an internal heat higher than the previous best evaluation from Pioneer/Voyager[24]. With the new updated value of internal heat, the current evolutionary theories and models of Jupiter should be revisited.

The significant wavelength-dependent nature of Jupiter's reflected solar radiance not only provides the best analyses of Jupiter's radiant energy budget and internal heat but also suggests that those quantities should be re-examined for other giant planets in our solar system. Previous analyses of the radiant energy budgets of Saturn, Uranus, and Neptune are based on observations with limitations in viewing angle and wavelength and the wavelength-dependent aspect was not well considered. The Cassini on-orbit observations of Saturn are even better than its fly-by observations of Jupiter in the coverage of viewing angle. In addition, Saturn's on-orbit observations (~13 years) are much longer than Jupiter's fly-by observations (~0.5 year). Therefore, the Cassini observations can be used to precisely measure Saturn's radiant energy budget and explore its temporal variations[25,34]. The energy balance of Uranus should also be re-examined. Estimates based on Voyager observations suggest there is no detectable internal heat on Uranus[35,36]. But detailed modeling[37] suggests that the internal heat on Uranus should be substantially larger than the estimates from the analyses based on the Voyager observations. Better observations and analyses addressing the wavelength-dependent nature of Uranus's radiant energy budget will help resolve the discrepancy among the previous studies.

Finally, the complete data sets of Jupiter's geometric albedo, phase integral, and Bond albedo in the wavelength range of 0–4000 nm (i.e., the wavelength range with significant solar spectral irradiance) provide a solid basis for the exploration of the radiant energy budget and internal heat on the giant planets in extra-solar systems.

## Methods

**Theoretical methodology**. The methodology of computing the radiant energy budget for an astronomical body has been well described in previous studies[1,2]. In order to determine the radiant energy budget, we need to compute two energy terms: the emitted thermal energy and the absorbed solar energy. The method of computing the emitted thermal energy of the gas giants (i.e., Jupiter and Saturn) and the measurements with observations from the Cassini spacecraft are discussed in our previous studies[25,26]. Here, we introduce the methodology of computing the full-disk Bond albedo, the reflected solar power, and hence the absorbed solar power for an astronomical body.

The estimate of absorbed solar energy for an astronomical body is based on the measurements of absorbed solar energy per unit time over a unit area (i.e., the absorbed power with a unit of W m$^{-2}$) $P_{absorb}$, which can be expressed as

$$P_{absorb} = (1 - A)\,\pi\,S/D^2 \qquad (1)$$

where $A$ is the Bond albedo of an astronomical body, $\pi S$ is the solar constant at Earth, and $D$ is the distance of the planet from the Sun in astronomical units (1 AU = 149.6 × 10$^9$ m). The only unknown variable in Eq. (1) is the Bond albedo. The Bond albedo $A$ is defined as the ratio between the reflected (or scattered) solar radiation $P_{reflect}$ and the incident solar radiation $P_{incident}$. The reflected solar radiation $P_{reflect}$ is a measure of the solar radiance reflected from all phase angles and all wavelengths, and the incident solar radiation $P_{incident}$ is determined immediately once the distance between a planet and the Sun is known ($P_{incident} = \pi S/D^2$). Assuming that the reflected solar radiance is independent of azimuth angle, the reflected solar radiation can be written as

$$P_{reflect} = 2\pi \int_{\lambda_1}^{\lambda_2} \int_0^{\pi} I_\lambda(\phi)\,\sin\phi\,d\phi\,d\lambda = 2\pi \sum_{\lambda_i} \sum_{\phi_j} I_{\lambda_i}(\phi_j)\sin\phi_j\,\Delta\phi\Delta\lambda \qquad (2)$$

where $\phi$ is the phase angle (i.e., the angle between the line from the Sun to the observed target and the line from the observer to the target), and $I_{\lambda i}(\phi_j)$ is the measured reflected/scattered solar radiance with wavelength $\lambda_i$ and phase angle $\phi_j$. The distributions of $I_{\lambda i}(\phi_j)$ over different wavelengths at a given phase angle are also called albedo spectra or reflectance spectra. Based on Eq. (2), the Bond albedo $A$ can be further converted into

$$A = \frac{P_{reflect}}{P_{incident}} = \frac{2}{S/D^2} \sum_{\lambda_i} \sum_{\phi_j} I_{\lambda_i}(\phi_j)\sin\phi_j\,\Delta\phi\,\Delta\lambda = \sum_{\lambda_i} \frac{I_{\lambda_i}(0)}{S/D^2} \left( 2\sum_{\phi_j} \frac{I_{\lambda_i}(\phi_j)}{I_{\lambda_i}(0)} \sin\phi_j\,\Delta\phi \right) \Delta\lambda$$
$$= \sum_{\lambda_i} A_0(\lambda_i)\,q(\lambda_i)\,\Delta\lambda = \sum_{\lambda_i} A_m(\lambda_i)\,\Delta\lambda$$
$$(3)$$

where $A_0(\lambda_i) = I_{\lambda_i}(0)/(S/D^2)$ is called the monochromatic geometric albedo,

### Table 1 Jupiter's radiant energies and internal heat

| Parameter | Power |
| --- | --- |
| Reflected solar radiation | 6.693 ± 0.160 W m$^{-2}$ |
| Absorbed solar radiation | 6.613 ± 0.160 W m$^{-2}$ |
| Emitted thermal radiation | 14.098 ± 0.031 W m$^{-2}$ |
| Internal heat | 7.485 ± 0.163 W m$^{-2}$ |

which is the monochromatic albedo at 0° phase angle for a given wavelength. The factor $q(\lambda_i)$, called the monochromatic phase integral, is the integral of the phase function $q(\lambda_i) = 2 \sum \left| I_{\lambda_i}(\phi_j) \sin \phi_j \Delta\phi / I_{\lambda_i}(0) \right|$. The monochromatic Bond albedo $A_m(\lambda_i)$, which is also called spherical albedo, is defined as $A_m(\lambda_i) = A_0(\lambda_i)q(\lambda_i)$. Once the monochromatic Bond albedo $A_m(\lambda_i)$ at each wavelength is measured, we can compute the Bond albedo $A$ for an astronomical body by Eq. (3).

There are two critical factors for computing the reflected solar power and hence the absorbed solar power by an astronomical body. First, the measurements of the reflected solar radiance should completely cover all wavelengths in which the solar spectral irradiance significantly contributes to the total solar power. Second, the measurements should completely cover all phase angles for the phase integral. For the Sun in our solar system, the solar spectral irradiance (SSI) over the wavelength range of 0–4000 nm contributes more than 99% of the total solar power. It is important to note that it is difficult to get the measurements with complete coverage of wavelength and phase angle, so some statistical methods are used to fill the observational gaps in wavelength and phase angle.

**Summary of observational data sets**. The theoretical methodology suggests that we need to know the monochromatic Bond albedo $A_m(\lambda_i)$ at all wavelengths in which the solar spectral irradiance significantly contributes to the total solar power in order to compute Jupiter's reflected solar power and hence the reflected/absorbed solar power. At each wavelength, the monochromatic Bond albedo $A_m(\lambda_i)$ depends on the monochromatic geometric albedo $A_0(\lambda_i) = I_{\lambda_i}(0)/S/D^2$ and the monochromatic phase integral $q(\lambda_i) = 2 \sum_{\phi_j} \left[ I_{\lambda_i}\left(\phi_j\right) \sin\phi_j \, \Delta\phi / I_{\lambda_i}(0) \right]$ (Eq. (3)).

The phase integral is determined by the variation of Jupiter's reflected radiance $\left( I_{\lambda_i}\left(\phi_i\right) \right)$ with phase angle (i.e., light curve or phase function). In general, the geometric albedo spectra (phase angle = 0°) are difficult to measure. So, here, we use the phase functions to determine the geometric albedo spectra from the available reflectance spectra at other phase angles.

The total solar power is mainly concentrated in the wavelengths range of 0–4000 nm (>99%), but Cassini observations do not cover this entire wavelength range. Therefore, we need other data to fill the observational gaps in wavelength. Besides supplementing the Cassini measurements, these other observations will also help validate the Cassini observations. Finally, we need to know the total solar power at Jupiter (i.e., solar constant at Jupiter) to compute the absorbed solar power (Eq. (1)) from the known Bond albedo. The total solar power can be computed from the integration of the SSI over the wavelength range of 0–4000 nm, so we need the measurements of SSI in the Cassini epoch. The data sets from the Cassini observations[28,29] and other observations[38–42] used in this study are summarized in Supplementary Table 1 and further introduced in the following section.

**Data sets of the SSI**. The SSI over the wavelength range of 0–4000 nm is used to compute the reflected/absorbed solar power with the measured Bond albedo at each wavelength. The total solar power varies on the order of magnitude ~0.1% on the time scale of decades[43,44], but the SSI at some wavelengths can vary with a much larger magnitude. Therefore, it is better to use the SSI in 2000–2001 to be consistent the observational time of the Cassini Jupiter-flyby mission (October 2000–March 2001). We attempt to get the SSI (0–4000 nm) in 2000–2001, or if unavailable at some wavelengths, at the closest possible times.

We build the SSI over the wavelengths 0–4000 nm from different data sets (Supplementary Fig. 1). We find the SSI over the wavelengths 120–420 nm in 2000–2001 from the Upper Atmosphere Research Satellite (https://uars.gsfc.nasa.gov/), but we cannot find the SSI in 2000–2001 in other wavelengths. Instead, we find the SSI over 0–120 nm in 2002 from the solar extreme ultraviolet radiation experiment (the observations began in 2002) (http://lasp.colorado.edu/home/see/) and the SSI over 420–2410 nm in 2003 from the solar radiation and climate experiment (the observations began in 2003) (http://lasp.colorado.edu/home/sorce/). In addition, the climatological SSI for the relatively long wavelengths over 2410–4000 nm comes from the American Society for Testing and Materials (http://rredc.nrel.gov/solar/spectra/am1.5/). Then we scale the SSI at Earth to the SSI at Jupiter by the inverse square of the ratio of their respective distances from the Sun. The distance between the Sun and Jupiter is averaged over the observational period of the Cassini Jupiter-flyby mission (October 2000–March 2001), which is ~5.039 Astronomical Units. The resulting SSI at Jupiter is used in the calibration of the Cassini observations and hence the measurements of Jupiter's full-disk albedo.

**The Cassini ISS/VIMS observations and data processing**. The Cassini spacecraft conducted Jupiter-flyby observations from October 2000 to March 2001. There were 12 scientific instruments working together to explore Jupiter's atmosphere, surface and interior. In this study, we measure Jupiter's reflected solar radiance based on the observations from two Cassini instruments: the ISS and the VIMS. The previous best estimate of Jupiter's radiant energy budget and internal heat[24] is mainly based on the observations from the Pioneer and Voyager spacecraft. The Charged-Coupled Device in the camera systems (e.g., ISS) of the Cassini spacecraft[28,29] is much more advanced than the old Vidicons in the Pioneer/Voyager spacecraft[45,46]. The radiance calibration of Cassini ISS and VIMS is

significantly improved over the calibration of Pioneer and Voyager imaging instruments. In addition, the coverage of wavelength and phase angle is much better in the Cassini observations than in the Pioneer/Voyager observations.

The ISS is a two-dimensional imaging device on the Cassini spacecraft[1]. The characteristics of the instrument ISS and the related data processing (e.g., calibrating and navigating) were described in previous studies[29,47]. The ISS images are taken from both of the narrow-angle camera and wide-angle camera. The ISS cameras have multiple filters (i.e., wavelengths) ranging from the ultraviolet to the near infrared. Here, we mainly use the 12 filters with central wavelengths ranging from 264 to 939 nm (Supplementary Table 2).

To compute Jupiter's full-disk albedo, we search the complete Cassini/ISS data set of the Jupiter-flyby observations (https://pds-imaging.jpl.nasa.gov/volumes/iss.html) for Jupiter's global images. We note that the global images with spatial resolution worse than 5000 km/pixel cannot resolve Jupiter very well, so only the global images with spatial resolution better than 5000 km/pixel are used in this study. The selected ISS global images include phase angles from ~ 0° to ~140° (i.e., the largest phase angle for the Cassini spacecraft during its Jupiter-flyby mission), which are the best among all available observations. The raw global images of Jupiter have observational gaps in phase angle and the observational gaps vary with the ISS filters. In order to increase the coverage of phase angle for the ISS observations, we also build Jupiter's global images based on some quasi-simultaneous hemispheric and quarterly images recorded by the ISS. Examples of making global images from quarterly images taken in a $2 \times 2$ mosaic are shown in Supplementary Fig. 2. There are still observational gaps in phase angle even with the new global images built from partial images, so we have to use interpolation/extrapolation to fill the observational gaps (see the following section).

Calibrating the ISS-recorded digital number of brightness to the radiance is critical for our measurements of Jupiter's full-disk Bond albedo. The ISS multi-filter images are calibrated by the Cassini ISS CALibration software (CISSCAL)[48]. The latest version (3.8) of CISSCAL (https://pds-imaging.jpl.nasa.gov/data/cassini/cassini_orbiter/coiss_0011_v3/) is used to calibrate the ISS images and output the images with a unit of absolute radiance (see an example in Supplementary Fig. 3). Then the absolute radiance at each pixel of the global images of Jupiter is multiplied by the area of the pixel, and summed over all pixels in the global images to get the observed full-disk reflected solar radiance. At the same time, the reference SSI (Supplementary Fig. 1) is scaled by the distance between Jupiter and Sun and multiplied by the total area of Jupiter to get the reference full-disk solar radiance. Then the ratio between the observed full-disk solar radiance and the reference full-disk solar radiance is taken as the full-disk albedo (i.e., I/F).

The Cassini ISS observations have the best coverage of phase angle, but the observations focus on 12 wavelengths only (Supplementary Table 2). On the other hand, the Cassini VIMS instrument[28] is a spectral camera that essentially takes images in 352 continuous channels between ~350 and 5100 nm. The channel spacing in wavelength averages 7.3 nm for channels 1–96 (VIMS–VIS) and 16.6 nm for channels 97–352 (VIMS–IR). The VIMS is designed to measure scattered and emitted light from surfaces and atmospheres with emphasis on both the spectral domain and spatial resolution[28]. The global VIMS images at different wavelengths are well calibrated by the VIMS Operations Team[28,49]. The detailed description of the VIMS calibration is introduced by the VIMS team report (please see https://pds-atmospheres.nmsu.edu/data_and_services/atmospheres_data/Cassini/logs/vims-radiometric-calibration-pds-2016-v1.20.pdf).

Examples of the calibrated global images recorded by the VIMS are displayed in Supplementary Fig. 4. Based on the incident angle, we divide the global image into day-side and night-side parts. The radiance recorded by the night-side part is mainly from thermal emission, and the radiance recorded by the day-side part includes both reflected solar radiance and thermal emission. In order to precisely measure the reflected solar radiance and hence full-disk albedo, we subtract the night-side thermal emission from the day-side radiance and average over the full disk of Jupiter, even though the night-side emission is generally much smaller than the day-side radiance in the wavelength range of 350–4000 nm.

The spatial resolution is generally much worse for the VIMS image than for the ISS images. As we discussed before, global images with spatial resolution worse than 5000 km/pixel cannot resolve Jupiter very well. So only VIMS images with spatial resolution better than 5000 km/pixel are used in this study. In addition, the coverage of phase angle is worse for the VIMS observations than for the ISS observations. We searched the complete data set of the VIMS observations (https://pds-imaging.jpl.nasa.gov/volumes/vims.html) and found high-quality global observations at seven phase angles (i.e., 10.3°, 14.9°, 63.3°, 76.6°, 88. 1°, 88.2°, and 110.2°) only (see Supplementary Fig. 5).

The Cassini spacecraft has one more imaging system—the Ultraviolet Imaging Spectrograph Subsystem (UVIS)[50], which observes Jupiter in wavelengths (56–190 nm) shorter than the ISS and VIMS. The SSI in the UVIS wavelengths (56–190 nm) occupies only ~0.13% of the total solar power. In addition, there are no existing processed UVIS spectra publically available in the Planetary Data System to our knowledge. Instead, we find the available ultraviolet spectra from other observations (see Supplementary Table 1). Therefore, the UVIS observations are not utilized in this study.

**Available albedo spectra from other observations**. As we discussed in section "Summary of observational data sets", the observations from the missions different

from Cassini can help to validate the Cassini measurements and fill the observational gaps in wavelengths. There are many observations of Jupiter's albedo spectra in different wavelengths. Here, we select observations from the European Southern Observatory (ESO)[39,40], one of the best observations of Jupiter's albedo spectra, which covers the large fraction of visible bands from ~310 to 1050 nm with very high spectral resolution (~0.4–1 nm) at phase angles 9.8° and 6.8°. In addition, the geometric albedo spectra in the ultraviolet wavelengths (see Supplementary Table 1), which are shorter than the shortest wavelength of the Cassini ISS observations (i.e., 266 nm), are used in this study.

Supplementary Fig. 5 shows the Cassini VIMS albedo spectra and the albedo spectra from other observations, which suggests a consistency of spectra shape between the VIMS observations and other observations over their overlap wavelengths (350–1050 nm). The magnitude of albedo spectra is different between the VIMS observations and other observations because they have different phase angles. The observations from ESO will also be used to validate the Cassini observations, which are discussed in the following Supplementary Information. Finally, the ESO albedo spectra[40] at 6.8° in the wavelength range of 300–1050 nm will be used to derive the albedo spectra at other phase angles based on the phase functions at the 12 wavelengths recorded by the Cassini ISS observations.

**Validation of the ISS and VIMS observations**. In this section, we use Jupiter's albedo at some particular phase angles, which come from the other observations introduced above (Supplementary Fig. 5), to validate the measurements by the Cassini ISS and VIMS. Figure S6 shows the phase functions (i.e., light curves) of Jupiter's full-disk albedo at the 12 wavelengths covered by the Cassini ISS. In addition, the Cassini VIMS observations (~350–4000 nm) in the ISS 12 wavelengths are selected for comparison. Finally, the observations from these missions different from Cassini (Supplementary Table 1 and Supplementary Fig. 5) are added for comparison over the ISS 12 wavelengths. Supplementary Fig. 6 demonstrates basic consistency between the Cassini observations and other observations.

We also conduct a comparison of Jupiter's full-disk albedo at given phase angles among different data sets. The ESO observations in 1993 have a phase angle ~9.8°, which is close to the phase angle ~10.3° of the Cassini/VIMS observations. In addition, there are some available observations with phase angles between 9.8° and 10.3° from the Cassini/ISS observations. We first average these available ISS observations over the range of phase angle 9.8°–10.3°. Then we compare Jupiter's full-disk albedo among the ESO, the Cassini/VIMS, and the Cassini/ISS, as shown in Supplementary Fig. 7. The ESO observations are well calibrated[39,40], and hence we take them as a standard reference. Supplementary Fig. 7 suggests that the Cassini/ISS results are generally larger than the ESO results except for a couple of wavelengths at the MT2 (728 nm) and MT3 (890 nm) filters. Conversely, the Cassini/VIMS results are generally smaller than the ESO results except for some wavelengths with absorptions (e.g., 619, 728, 890 nm). The different directions of the discrepancy from the ESO results between the Cassini ISS (generally larger than the ESO) and the Cassini VIMS (generally smaller than the ESO) implies that the differences between the ESO results and the Cassini results are unlikely related to the possible temporal variations of Jupiter's albedo (see section "Possible temporal variations of Jupiter's energy budget").

Supplementary Fig. 7 suggests that the Cassini ISS and VIMS results generally overestimate and underestimate Jupiter's full-disk albedo, respectively. Some discrepancies between the ESO results and the Cassini results can be explained. For example, the discrepancy between the Cassini/VIMS result and the ESO result at the MT3 wavelength (890 nm) is probably related to the low spectral resolution of the VIMS observations (~7.3 nm at the 890 wavelength). The methane-absorption band MT3 is centered on ~887.8 nm with full widths at half maximum ~3 nm[39,40]. The VIMS observations with a low spectral resolution cannot resolve the strong methane-absorption band at the MT3 wavelength very well. The EOS observations with a high spectral resolution (~0.4–1 nm) can resolve the absorption band the MT3 wavelength much better. That probably explains why the VIMS result is different from the ESO result at the MT3 wavelength. For most discrepancies between the Cassini results and the EOS results, we do not have good explanations at this moment. The calibration of the Cassini ISS/VIMS observations is complicated[48,49]. The team members of the Cassini ISS and VIMS, which include some authors of this paper (e.g., Robert West and Kevin Baines), are still working on the calibration of the Cassini data sets.

The largest discrepancies between the Cassini results and the ESO results can reach ~10% (e.g., the discrepancy between the Cassini/ISS and the EOS at the ISS GRN wavelength ~568–569 nm). But the average discrepancies between the Cassini results and ESO results are ~1.3% and ~2.2% for the Cassini ISS and VIMS, respectively. The discrepancies between the Cassini observations and the ESO observations are further discussed with the standard deviation for these phase angles with multiple measurements from the Cassini, the ESO, and other sources (see Supplementary Table 1). The ratio of standard deviation over mean for these phase angles with multiple measurements is shown in Supplementary Fig. 8, which suggests that the ratio is basically less than a few percent except for some measurements at the wavelengths of GRN, MT2, and MT3 filters of Cassini ISS. Even for these ISS filters/wavelengths with large ratios (i.e., GRN, MT2, and MT3), the average ratio over all phase angles is still small (<5%). It should be mentioned that the discrepancies of Jupiter's full-disk albedo among different data sets and the

related standard deviations are considered in the uncertainty estimate of the results presented in this study (see section "Uncertainty estimate").

**Filling observational gaps in phase angle**. There are observational gaps in phase angle for the Cassini ISS and VIMS observations (Supplementary Fig. 6). So we have to fill the observational gaps after validating the Cassini ISS and VIMS measurements. We use the least-squares method[30] to fit the observed phase functions, then the fitting results are used to fill the observational gaps. Different functions were used to fit the phase functions of Jupiter's full-disk albedo[31,51,52]. Among them, the two-term Henyey–Greenstein (H–G) function $P(A_{HG}, g_1, g_2, f, \phi)$, which is expressed as below, is generally used.

$$P(A_{HG}, g_1, g_2, f, \phi) = A_{HG} \cdot (f P_{HG}(g_1, \phi) + (1-f) P_{HG}(g_2, \phi)) \quad (4)$$

where $A_{HG}$ is the coefficient to match the amplitude of the observed phase function. The term $P_{HG}(g, \phi)$ represents both forward (with a factor $g_1$ and $g_1 \in [0,1]$) and backward (with a factor $g_2$ and $g_2 \in [-1, 0]$) scattering lobes, respectively. The factor $f$ ($f \in [0,1]$) stands for the fraction of the forward versus backward scattering. The term $P_{HG}(g, \phi)$ ($g$ can be $g_1$ or $g_2$ and $\phi$ is phase angle) has the following form:

$$P_{HG}(g, \phi) = (1 - g^2)/(1 + g^2 + 2g \cdot \cos\phi)^{3/2} \quad (5)$$

We test the fitting of the H–G function with the least-squares method for the Cassini ISS observations at the second continuum band filter (CB2). The CB2 is the filter at which the largest number of the ISS observations were obtained, so the CB2 observations have the best coverage of phase angle. We use two methods to fit the CB2 observations with the H–G function: (1) with specifying the ranges of parameters $f$ ($f \in [0,1]$), $g_1$ ($g_1 \in [0,1]$), $g_2$ ($g_2 \in [-1, 0]$), and (2) without specifying the ranges of parameters $f$, $g_1$, and $g_1$. In addition, we try a simple second-order polynomial function as below to fit the CB2 observations.

$$P(\phi) = c_1 \phi^2 + c_2 \phi + c_3 \quad (6)$$

where $P(\phi)$ is the fitting phase function. The parameters $c_1$, $c_2$, and $c_3$ are fitting coefficients to match the observations with the least-squares method.

The fitting results are shown in Supplementary Fig. 9, which suggests the H–G function without specifying parameter range is better than the H–G function with specifying parameter range. However, both H–G fittings are worse than the fitting with a polynomial function. It suggests that the H–G function, which is based on single scattering, probably does not work perfectly for the phase function of Jupiter's full-disk albedo. Therefore, we use the polynomial function (Eq. (6)) to fit the phase function of Jupiter's albedo, and hence fill the observational gaps in phase angle for the observations obtained by the Cassini ISS 12 filters.

Supplementary Fig. 10 displays the fitting results for the Cassini ISS observations, which suggest that the simple polynomial function works well for all 12 filters. Supplementary Fig. 11 shows the fitting residual ratio, which is the ratio of the fitting residual (i.e., fitting value subtracts observational value) over the corresponding observation. The fitting residual ratio is less than 10% for all 12 filters, which demonstrates again that the polynomial-function fitting works for fitting the ISS observations and hence filling the observational gaps.

Compared to the Cassini ISS observations, the coverage of phase angle for the high-quality Cassini VIMS observations is limited. As discussed in Supplementary Notes 3 and 4, only seven high-quality VIMS observations with spatial resolution better than 5000 km/pixel are selected for the measurements of Jupiter's full-disk albedo. The seven VIMS observations were recorded at seven different phase angles (10.3°, 14.9°, 63.3°, 76.6°, 88. 1°, 88.2°, and 110.2°, see Supplementary Fig. 5). We first test if the VIMS observations at the seven phase angles are good enough for the fitting with the least-squares method.

Supplementary Fig. 12 shows the comparison of fitting with the polynomial function (Eq. (6)) between the ISS observations and the VIMS observations over the overlap wavelengths between them. The comparison suggests that the fitting results are basically consistent between the ISS observations and the VIMS observations. Therefore, the polynomial-function fitting also works for the VIMS observations and we use it to fit the phase angles at all VIMS wavelengths (350–4000 nm). Supplementary Figs. 13, 14 display the fitting results and fitting residual ratio, respectively. The fitting residual ratio (i.e., the ratio between the fitting residual and the VIMS observational value) at the seven VIMS phase angles suggests that the fitting residual ratio is generally small (<5%) in the relatively short wavelengths (i.e., ~350–1000 nm) and the fitting residual ratio is generally large (~10–15%) in the relatively long wavelengths (i.e., 1000–4000 nm). The large fitting residual ratio in the relatively long wavelengths is mainly due to the generally small full-disk albedo in these long wavelengths (Supplementary Fig. 13). The large fitting residual in the relatively long wavelengths does not significantly affect our measurements of Jupiter's total reflected solar power because the SSI (Supplementary Fig. 1) is much smaller in the long wavelengths (i.e., 1000–4000 nm) than in the short wavelengths (i.e., 350–1000 nm).

**Filling observational gaps in wavelength**. The VIMS observations (350–4000 nm) (Supplementary Fig. 13) do not cover the wavelengths shorter than 350 nm. In order to compute Jupiter's full-disk albedo, we need Jupiter's albedo in the

complete two-dimensional (2-D) domain of phase angle and wavelength. Therefore, we need the measurements of Jupiter's albedo in the wavelength range of 0–350 nm with complete coverage of phase angle. Supplementary Fig. 5 shows that the magnitude of Jupiter's albedo spectra change with phase angle but the spectral structure and shape basically stay constant. It means that we can use Jupiter's albedo spectra at the available phase angles (Supplementary Fig. 5) to derive the albedo spectra over the whole range of phase angle (i.e., 0°–180°). Based on the least-squares technique discussed in Supplementary Note 6, we first determine the complete phase functions of Jupiter's albedo over the whole range of phase angle for these wavelengths of the Cassini ISS 12 filters. Then, we use the complete phase functions (i.e., distribution over 0°–180°) at the Cassini ISS 12 wavelengths to derive the phase functions at all wavelengths from 264 nm (i.e., the shortest wavelength of the ISS observations) to 939 nm (i.e., the longest wavelength of the ISS observations) by referring to the spectral shape at a phase angle 6.8° from Karkoschka[40] (see Supplementary Fig. 5). By this way, the derived albedo spectra in the wavelength range of 264–939 nm keep the structure/shape of the available 6.8° spectra from Karkoschka[40] over the whole range of phase angle (0°–180°), as suggested by Supplementary Fig. 5.

For the wavelengths shorter than 264 nm (i.e., 0–264 nm), we first extrapolate the complete phase function at 264 nm from the Cassini ISS observations to the wavelength range of 0–264 nm. Then we interpolate/extrapolate the available geometric albedo spectra from International Ultraviolet Explorer (125–195 nm)[38] and Aerobee rocket (210–300 nm)[40,41], which are discussed in Supplementary Table 1 and Supplementary Fig. 5, to the wavelength range of 0–264 nm. Finally, the extrapolated phase function in each wavelength in the wavelength range of 0–264 nm is combined with the available geometric albedo (i.e., 0° phase angle) at the same wavelength to derive Jupiter's albedo over the whole range of phase angle at each wavelength, as we did for the wavelength range of 264–3939 nm (see Supplementary Fig. 15).

The derived Jupiter's albedo in the wavelength range of 0–939 nm (Supplementary Fig. 15) not only helps fill the observational gaps in wavelength for these wavelengths (0–350 nm) outside the VIMS observations but also strengthens the validation of the VIMS fitting results (Supplementary Fig. 13) for the overlap wavelengths (350–939 nm) between the VIMS and the ISS. Supplementary Fig. 16 shows the comparison of Jupiter's albedo between the ISS derived results (Supplementary Fig. 15) and the VIMS fitting results (Supplementary Fig. 13) for the overlap wavelengths (350–939 nm). The comparison shows that the results are basically consistent between ISS (panel A) and VIMS (panel B) in the relatively low phase angles (0°–90°) except for wavelengths around the ISS MT2 and MT3 filters. The large discrepancy between ISS and VIMS around the wavelengths of MT3 filter occurs because the VIMS observations do not resolve the fine spectral structures around the MT3 filter very well due to the low spectral resolution. Panel C of Supplementary Fig. 16 also shows that the difference ratio (i.e., the ratio of the difference between the ISS and VIMS over their mean) gets larger (>10%) in the high phase angles (i.e., 90°–180°), and it is because Jupiter's albedo becomes very small (<0.1) with phase angles larger than 90°. The large discrepancy in high phase angles between ISS and VIMS basically does not affect our measurements of Jupiter's total reflected solar power because the contribution is much less from the small albedo in the high phase angles (i.e., >90°) than from the large albedo in the low phase angles (<90°).

Supplementary Fig. 16 suggests roughly consistent results between the ISS derived albedo and the VIMS fitting albedo. In this study, we use the ISS derived results for the overlap wavelengths (i.e., 350–939 nm) between the ISS and the VIMS because: (1) the coverage of phase angle is much better for ISS observations than for the VIMS observations so that the fitting phase functions are more precise for the ISS observations than for the VIMS observations; and (2) the spectra from the ESO (Supplementary Fig. 5), which are used in the ISS derived albedo, have a very high spectra resolution ~0.4–1 nm. Such a spectral resolution is much better than the spectral resolution of the VIMS observations (~4–24 nm), so that some fine spectral structures (e.g., the strong methane-absorption band at the ISS MT3 filter) can be better resolved.

**Uncertainty estimate.** After filling the observational gaps in phase angle and wavelength, we have Jupiter's full-disk albedo in the complete domain of phase angle and wavelength. Therefore, we can compute the total reflected solar power by Jupiter, which is discussed in the main text. Here, we discuss the uncertainties (i.e., error bars) in computing Jupiter's Bond albedo and the solar power reflected by Jupiter. There are three uncertainties considered in our measurements of Jupiter's Bond albedo and reflected solar power: (1) the uncertainty from calibrating the Cassini ISS and VIMS images; (2) the uncertainty in filling the observational gaps with the least-squares fitting; and (3) the standard deviation of multiple measurements including the Cassini ISS/VIMS observations and other observations. The three uncertainties are combined together for analyzing the total uncertainty in each point of albedo in the 2-D domain of phase angle and wavelength (Fig. 2 in the main text).

For uncertainty (1) (i.e., calibration of the Cassini ISS/VIMS observations), we mainly refer to previous studies[28,29,48,49]. The calibration uncertainty of the Cassini ISS observations varies with filters, viewing geometry, and observed object. A previous study[48] suggests that the calibration error of the Cassini ISS images is approximately a few percent of the calibrated radiance. Here, we simply assume

that the calibration error is 5% of the ISS radiance. For the calibration uncertainty of the Cassini VIMS observations, we refer to a previous study[49], in which the absolute uncertainty was estimated to be 5–10% of the recorded radiance. In this study, we use the average error (i.e., 7.5%) for the calibration uncertainty for the VIMS observations. For uncertainty (2) (i.e., filling observational gaps with fitting), we use the fitting residual, which exists in these points with available Cassini observations, to estimate the uncertainty in the observational gaps. For uncertainty (3) (i.e., standard deviation of multiple measurements), it only exists in these points with multiple measurements (see Supplementary Fig. 8). We interpolate and extrapolate the uncertainties (2) and (3) from the points with observations to the observational gaps.

Supplementary Fig. 17 shows the comparison between the uncertainty (2) and the uncertainty (3), which suggests that they are comparable for most phase angles in the overlap wavelengths between the ISS and the VIMS. Supplementary Fig. S17 further suggests that the uncertainties (2) and (3) have a magnitude of a few percent, which is also comparable to the uncertainty from the Cassini calibration (i.e., uncertainty (1)). It should be mentioned that the standard deviation of multiple measurements (Supplementary Fig. 8), which is considered as the uncertainty (3), would include possible temporal variations of Jupiter's albedo (see section "Possible temporal variations of Jupiter's energy budget"). The three uncertainties, which are discussed above, are combined together for the error bars shown in Fig. 3 in the main text. The direct measurements of the monochromatic geometric albedo, which is the albedo at 0° phase angle, are relatively limited. We basically extrapolate the uncertainty at the phase angles with available observations for the error bars of the monochromatic geometric albedo (panel A of Fig. 3 in main text). In estimating the uncertainty of the monochromatic phase integral (pane B of Fig. 3 in main text), the formula of error propagation is used. Based on Eq. (3) in the main text, we have the expression of the monochromatic phase integral as below.

$$q(\lambda_i) = 2 \sum_{\phi_j} \frac{I_{\lambda_i}(\phi_j)}{I_{\lambda_i}(0)} \sin\phi_j \, \Delta\phi = \sum_j c_j I_{\lambda_i}(\phi_j) \tag{7}$$

where $c_j = 2 \sin\phi_j \, \Delta\phi / I_{\lambda_i}(0)$ is the coefficients at different phase angles. Then the square of error of phase integral $\sigma^2(q)$ can be expressed as the sum of the square of error of the measurements in different phase angles ($\sigma^2(I(\phi_j))$).

$$\sigma^2(q) = \sum_j c_j^2 \sigma^2(I(\phi_j)) \tag{8}$$

In estimating the uncertainty of the monochromatic Bond albedo (panel C of Fig. 3 in main text), we use the following formula for the error propagation. At each wavelength, the monochromatic Bond albedo $A_m(\lambda_i)$ can be expressed as the product of the monochromatic geometric albedo and phase integral (i.e., $A_m(\lambda_i) = A_0(\lambda_i)q(\lambda_i)$). Then we have the following equation for the error of the monochromatic Bond albedo ($\sigma(A_m)$) based on the errors of the monochromatic geometric albedo and phase integral ($\sigma(A_0)$ and $\sigma(q)$).

$$\frac{\sigma^2(A_m)}{A_m^2} = \frac{\sigma^2(A_0)}{A_0^2} + \frac{\sigma^2(q)}{q^2} \tag{9}$$

Based on the distribution of error of the monochromatic Bond albedo (Panel C of Fig. 3 in main text), we can compute the wavelength-average Bond albedo (i.e., Jupiter's Bond albedo $A$) by weighting the monochromatic Bond albedo with the SSI in 2000–2001 (Supplementary Fig. 1). The Bond albedo over wavelength is taken to the weighted mean of the monochromatic Bond albedos. Therefore, the error of Jupiter's Bond albedo can be estimated in a similar way by estimating the error of the phase integral (Eq. (8)).

**Comparison between the Cassini and previous measurements.** The radiant energy budget of Jupiter has been studied for a long time[24,53–64]. The study conducted by Hanel et al.[24] is based on the observations from the Pioneer and Voyager missions, which are the best observations before the Cassini epoch. Therefore, the results provided by Hanel et al.[24] are recognized as the best estimates of Jupiter's radiant energy budget and widely used in the community of planetary science. Here, we mainly compare the data and the related studies between Voyager/Pioneer[24] and Cassini (this study).

Jupiter's radiant energy budget is mainly determined by the emitted thermal power and absorbed solar power (see Methods section "Theoretical methodology"). Before the Cassini epoch, the best infrared observations for measuring Jupiter's emitted power come from Pioneer[62] and Voyager[24,64]. Compared to the infrared instruments of Pioneer/Voyager[65], Cassini's infrared instrument has extended far-infrared coverage and better spectral resolution[25,27]. In addition, the 5-μm thermal emission, which is important for Jupiter's total emitted power, was not addressed well in the previous studies[24,64] due to limited observations. Therefore, one of our previous studies[26], which is based on the Cassini thermal observations and addresses the 5-μm thermal emission, is better than the previous best study of Jupiter's emitted power[24].

The previous best estimate of Jupiter's Bond albedo and hence absorbed solar power is mainly based on the visible observations of Pioneer and Voyager[24,31], even though some old measurements[56] before the Pioneer/Voyager epochs are used. The

visible observations conducted by the imaging photopolarimeter (IPP)[66] aboard Pioneer have two channels at blue (390–490 nm) and red (580–700 nm). The Pioneer/IPP global observations of Jupiter, which were used to examine the phase function/integral of Jupiter's full-disk albedo at red and blue channels[31], have discrete phase angles of 12°, 23°, 34°, 109°, 120°, 127°, and 150°. The visible observations conducted by the Radiometer of the Voyager IRIS instrument[65] have a single channel from 300 to 1900 nm. The Voyager/Radiometer observations, which were used to explore Jupiter's phase integral and Bond albedo[24], have a single phase angle ~24.7°. As a contrast, the visible observations conducted by the Cassini ISS have 12 channels from ultraviolet to near infrared (see Supplementary Table 2). In addition, the Cassini VIMS observations also provide visible and near infrared observations with quasi-continuous wavelengths. Therefore, the spectral coverage is much better in the Cassini observations than in the Pioneer/Voyager observations. The Cassini visible observations, especially the ISS observations, have pretty good coverage of phase angle from ~0° to 144° (see Supplementary Fig. 6). Such a good coverage of phase angle is important to examine the phase maximal/integral of Jupiter's full-disk albedo. More importantly, the Cassini observations at a phase angle ~0°, which are lacking in the Pioneer/Voyager observations, are important for determining Jupiter's geometric albedo.

The above discussion suggests that the Cassini visible and infrared observations are better than the corresponding observations from Pioneer/Voyager. Therefore, we think the results based on the Cassini observations are more robust than the results from previous studies. Here we provide a detailed comparison of the radiant energy components between the results from the Cassini observations (this study) and the previous best estimates from the Pioneer/Voyager observations[24].

First, we do not find large difference of global-average emitted power between our study from Cassini[26] (~14.098 ± 0.031 W m$^{-2}$) and the previous results from Pioneer/Voyager (13.8 ± 1.4 and 13.59 ± 0.14 W m$^{-2}$ for Pioneer and Voyager, respectively)[24,62], even though the meridional profile of emitted power significantly changed from the Pioneer/Voyager epochs to the Cassini epoch[26]. The other energy component—absorbed solar power is mainly determined by the Bond albedo. Regarding to the Bond albedo, we find the large difference between our study (0.503 ± 0.012) and the previous result (0.343 ± 0.032) by Hanel et al.[24]. The Bond albedo is further determined by two factors: geometric albedo and phase integral. The best value of phase integral (~1.2–1.3) based on the Pioneer observations at red and blue filters[31] is basically consistent with our Cassini result (~1.1–1.3) over the whole spectral range (see Fig. 3 in main text).

The significant difference of Bond albedo between our study and the previous best estimate[24] is mainly due to the large difference in geometric albedo between the previous study (~0.274 over 0.4–1.7 μm)[24] and this study (0.442 over 0.4–17 μm) (also see Fig. 3 in main text). Our analyses of Jupiter's geometric albedo based on the Cassini observations are better than previous estimates based on the Pioneer/Voyager observations for the two following reasons: (1) the Cassini observations at ~0°, which are lacking in the previous Pioneer/Voyager observations, can be used to measure the geometric albedo directly; and (2) our Cassini analyses are validated by the best ground-based observations[39,40]. It seems that the analysis of geometric albedo in the previous study by Hanel et al.[24] used some results from a study by Taylor[56], in which the geometric albedo was estimated to be ~0.28 over 0.34–1 μm. The study by Taylor[56] is based on the observations made with an old monochromator, which has very large uncertainties. In addition, the phase integral (~1.6) based on the old monochromator observations[56] has been proved to be wrong[31].

Therefore, we think that the best previous study[24] used the wrong geometrical albedo so that Jupiter's Bond albedo was seriously underestimated. However, it is still possible that there are temporal variations of Jupiter's Bond albedo and hence radiant energy budget from the Pioneer/Voyager epochs to the Cassini epoch, which are discussed in the following section.

**Possible temporal variations of Jupiter's energy budget.** The above discussion suggests that our results based on the Cassini observations are more robust than the results from previous studies. Here, we investigate possible temporal variations of Jupiter's radiant energy budget. Panel A of Supplementary Fig. 18 shows the difference of albedo spectra between 1993 and 1995, which is based on the previous studies by Karkoschka[39,40]. The two measurements in 1993 and 1995 have different phase angles, so the difference between 1993 and 1995 is at least partly due to the varying phase angle. Based on the phase functions from the Cassini observations, we derive the 1993 spectra from 9.8° (1993) to 6.8° (1995). Then we compare with the derived 1993 spectra at 6.8° phase angle with the observed 1995 spectra at the same phase angle. Panel A of Supplementary Fig. 18 suggests the difference is larger between the observed 1993 spectra (9.8°) and the observed 1995 spectra (6.8°) than between the derived 1993 spectra and the observed 1995 spectra with the same phase angle (6.8°). Panel B of Supplementary Fig. 18 further suggests that the original difference ratio between 1993 and 1995 significantly decreases to less than 5% for most wavelengths when we transform the 1993 spectra from 9.8° to 6.8°. Therefore, the temporal variation of Jupiter's full-disk albedo is probably less than a few percent from 1993 to 1995, even though the albedo in some specific areas of Jupiter's surface can vary greatly[67–70].

In addition to the short-term variation of Jupiter's full-disk albedo, we explore the possible long-term variations of Jupiter's full-disk albedo based on one of our previous studies[69]. The temporal variations of Jupiter's brightness in five visible and near-infrared bandpasses (U, B, V, R, and I filters with wavelengths 360, 436,

549, 700, and 900 nm, respectively) from 1963 to 2011, which are based on the observations from three sources of standardized wide-band photoelectric photometry for Jupiter, are discussed in our previous study[69]. These data sets used in the previous study[69] basically represent the long-term high-quality observations of Jupiter by wide-band filter photometry. One author of the previous study[69] and this study (R.W. Schmude Jr.) also recorded and processed the new data from 2011 to 2015. The observed astronomical magnitudes in the U, B, V, R, and I filters are first adjusted to a uniform distance of 1 AU by considering the distances between Sun and Jupiter and between Earth and Jupiter. The astronomical magnitudes are further converted to brightness values and averaged over the whole time period (1963–2015). We subtract the time-average value from all observed brightness values to get the anomaly brightness. Then the ratio between the anomaly brightness and time-average brightness is used to represent the anomaly percentage of brightness. Finally, the anomaly percentage of brightness at different phase angles is adjusted to the anomaly percentage of brightness at the phase angle 0° by a phase function[69]. The phase-angle adjustment is small because the maximal phase angle of all observations is ~11°. The adjusted anomaly percentage of brightness at 0° is used to represent the anomaly percentage of geometric albedo, which is shown in Supplementary Fig. 19.

Supplementary Fig. 19 suggests that Jupiter's geometric albedo is basically constant with time with the largest anomaly percentage ~15%. The average of the anomaly percentage of geometric albedo during the period of 1963–2015 is 3.7%, 3.7%, 2.7%, 2.9%, and 3.4% for U, B, V, R and I filters, respectively. The phase integral is relatively stable with time based on the comparison between the Pioneer result[34] and the Cassini result (this study). Therefore, we think the long-term temporal variations of Jupiter's Bond albedo are at most a few percent. Such possible temporal variations are much smaller than the 47% ((0.503–0.343)/0.343 ~47%) difference of Jupiter's Bond albedo between the best previous estimate[24] (0.343 ± 0.032) and this study (0.503 ± 0.012). So it is unlikely the 47% difference is due to temporal variations, and that supports our discussion in Supplementary Note 9 in which the previous estimate[24] is thought be seriously underestimate Jupiter's Bond albedo. It should be mentioned that the standard deviation of multiple measurements in different times (Supplementary Fig. 8), which includes the temporal variations of Jupiter's Bond albedo, have been considered in the uncertainty estimate of our measurements (see Methods section "Uncertainty estimate").

**Jupiter's disk area and global surface area.** The incident solar radiance is reflected/absorbed by the projected disk of Jupiter in the Sun's direction. The area of the disk can be expressed as $S_{inc} = \pi R_{Eq} R_{Po}$, where $R_{Eq}$ and $R_{PO}$ are the equatorial and polar radii, respectively. Jupiter's equatorial and polar radii at the 1-bar pressure level are 71,492 and 66,854 km, respectively[71]. Therefore, we have the area of Jupiter's imaginary disk absorbing/reflecting solar radiance as $S_{inc} = \pi R_{Eq} R_{Po} = 1.5015 \times 10^{10}$ km$^2$. The global surface area of Jupiter (i.e., an oblate planet) can be calculated as[72]

$$S_{glo} = 2\pi R_{Eq}^2 \left[ 1 + (1 - e^2)\tan h^{-1} e/e \right] \qquad (10)$$

where $e^2 = 1 - R_{Po}^2/R_{Eq}^2$. Based on Eq. (10), we have the global area of Jupiter as $6.03450 \times 10^{10}$ km$^2$. Therefore, the ratio between the area of the imaginary disk and the global area for Jupiter is $S_{inc}/S_{glo} = 0.2488$. Such a ratio is used to average the incident solar power over the global area of Jupiter, as discussed in the main text.

## Data availability
The original data recorded by the three Cassini instruments (CIRS, ISS, and VIMS) are publicly available and can be freely downloaded from the NASA Planetary Data System (https://pds.nasa.gov/). The data for the most important scientific results are presented in the article and the Supplementary Information. The processed data sets and the corresponding codes of Jupiter's radiant energy budget are available from L.L. upon reasonable request.

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

## Acknowledgments

The leading author (Liming Li) wants to thank one of his postdoc advisors, Dr. Barney J. Conrath, who encouraged him to conduct the studies of the radiant energy budgets of giant planets. In addition, we gratefully acknowledge the Cassini CIRS, ISS, and VIMS teams for recording the raw data sets. L.L. acknowledges the support from the NASA ROSES Cassini Data Analysis Program and Planetary Data Archiving, Restoration, and Tools Program. X.J. is supported by NASA grants NNX13AC04G, NNX13AK34G, and NNX16AG46G. S.P.-H. and A.S.-L. are supported by MINECO project AYA2015-65041-P (FEDER/EU) and Grupos Gobierno Vasco IT-765-13.

## Author contributions

L.L. performed the computations/analyses of Jupiter's radiant energy budget/internal heat and wrote the manuscript. X.J. processed the Cassini ISS data set and contributed to the computation. R.A.W., S.P.E., U.A.D., B.K. and C.C.P. contributed to the Cassini ISS data processing and calibrating. P.M.F. and K.H.B contributed to the Cassini VIMS data processing and calibrating. A.M. and R.W.S. Jr. provided long-term data sets of Jupiter's albedo. X.J., R.A.W., P.J.G., S.P.-H., A.S.-L., L.N.F., J.J.F., B.K., C.C.P., K.H.B., P.M.F., A.M., R.K.A., A.A.S., C.A.N., G.S.O., U.A.D., S.P.E. and R.W.S. Jr. contributed to the data analysis. All authors discussed the results and commented on the manuscript.

## Additional information

**Competing interests:** The authors declare no competing interests.

