## [Peer Review File · Nature Communications]

Reviewers' comments:

Reviewer #1 (Remarks to the Author):

This paper reports on a re-analysis of the observational data available for the reflectivity of Jupiter at different wavelengths and phase angles. This allows for a calculation of Jupiter's albedo, and from that its internal heat source. The conclusion of this work is that this internal heat source is nearly 40% higher than previous estimates. This is a surprising new result that has important consequences for models of Jupiter's evolution, and by extension, for gaseous exoplanets in general. It is of great interest to the community and should be published.

Because of the importance of this result, I would like to see some of the steps in the process made more robust. For example, the SSI in the UV can be quite variable. Some statement should be made in the supplementary material as to the size of this variation and its contribution to the overall uncertainty in Jupiter's albedo and its internal heat source.

In panel A of figure 17 in the supplementary material a comparison is made with the measurements of Karkoschka (1994, 1998). Although the agreement over most of the range is excellent, the difference between the 1993 (green) and 1995 (red) estimates are largest in the visible wavelengths where the solar irradiation is the strongest. Are there other measurements that could be compared with as well? Is the temporal change indeed limited to these values or could it have been larger in the past? Can the data be compared with Hubble observations or any other spacecraft observations? I would urge the authors to look for at least one additional albedo measurement in the literature for a more thorough comparison.

A small typo that caught my eye was in the supplementary material line 417 where instead of "production" it should say "product".

Reviewer #2 (Remarks to the Author):

In their paper, Li et al. provide updated estimates of Jupiter's internal heat, albedo and absorbed solar energy obtained from the analysis of (mainly) Cassini data acquired during the flyby of Jupiter in 2000-2001. These values are significantly different than that obtained from previous estimates (at the Pioneer and Voyager epoch) : Li et al derive a Bond albedo of 0.503 (compared to the previous estimate of 0.343) and consequently an internal heat flux of 7.485 W/m² (compared to 5.44 W/m²). These parameters are crucial to many studies : the history and evolution of the giant planets, their energy budget, their atmospheric circulation, etc. Hence, the topic of this paper is relevant to be published in Nature Communications, as it has a broad impact. The results look convincing and the paper is rather well written, although the supplementary material is a bit long. Several concluding remarks about the internal structure of Jupiter or application to exoplanet atmospheres are a bit far-stretched or need more justifications. General and specific comments are detailed below.

General comments :

Several times in the paper and supplementary material, the Cassini measurements are referred to as having a « better » or « more complete » phase coverage compared to Pioneer measurements, but this point deserves to be better documented or quantified. Is it possible to give the reader an idea of the phase angles and wavelengths covered by Pioneer measurements for instance ?

A related comment is about Figure 3 and its discussion lines 109-122 : is it possible to pinpoint the reason for the disagreement with the previously derived Bond albedo by Hanel et al ? For instance, is the geometric albedo in agreement with previous studies but not the phase integral, or the other way around ; or is this an effect of the wavelength coverage ?

Lines 164-168, regarding evolutionary and interior models : To my understanding, there are many arguments that suggest the existence of compositional gradients in giant planet interiors, leading to layered double diffusive convection (similar to the circulation in the Earth's oceans). To say that taking into account layered double diffusive convection is «potentially unnecessary » is a dangerous statement, as one cannot choose to discard a physical effect simply based on Li et al new results. Furthermore, lines 165-166, the statement that layered convection acts to « weaken the internal heat » is unclear. Indeed, layered double diffusive convection acts to slow the cooling of giant planet interiors and is used to explain, for instance, the excess of luminosity of Saturn (see for instance Leconte and Chabrier, Nature Geosciences 2013). Hence I do not understand how and why the revisited value of Jupiter's internal heat flux would make the layered convection theories obsolete.

Lines 166-168 and in the abstract, the authors state that with their revised value for Jupiter's internal heat, « a consistent evolutionary history for both Jupiter and Saturn [...] can finally be obtained » : again, I find that this statement is a bit far-stretched without a model to support it. Secondly, just afterwards (lines 169-172) the authors advocate for the revision of the value of internal heat for Saturn (and other giant planets). These two statements are in contradiction :if Saturn's internal heat is revisited and a very different value is obtained, as suggested, then new models would be needed to explain the difference between Jupiter and Saturn heat budget and evolution...

I also do not understand the relevance of the comparison to cool giant exoplanets in the last paragraph. After all, Saturn's albedo has been estimated to 0.34, and we can expect a great diversity of albedo in extrasolar cool giant planets depending on their composition, cloud and aerosol properties ; or the stellar type. The fact that Cahoy et al., 2010 failed to match Jupiter's albedo is not relevant, as the authors modeled a « Jupiter-like » exoplanet and did not fine-tuned their atmospheric model to match Jupiter's cloud and haze properties. Indeed, Cahoy et al clearly indicate that they do not take into account NH₄SH cloud but only a NH₃ cloud (which might not be relevant) ; that their model lack haze layers ; and that their geometric albedo spectrum is clearly at odds and too bright compared to Karkoschka et al, 1994 albedo spectrum at wavelengths shorter than 0.5 μ m. Hence it is not surprising that they derive a bright geometric albedo for their « Jupiter-like » planet. For these reasons, I recommend removing the reference to Cahoy et al., 2010.

Specific comments in Supplementary material :

Supp. Info 3 : A reference to Figure 1 is missing

Line 172 : « The Cassini spacecraft has one more instrument » : one more camera (or imaging system) instead ?

Figure S5 is very interesting but it is hard to see how well the VIMS and ESO data compare at a similar phase angle (about 10°). Actually, a comparison with the 12 channels of ISS at the same phase angle would be a great addition. Indeed, the following figures compare ISS and VIMS albedo versus phase angle at different wavelength, but the comparisons of the albedo spectra at a given phase angle would be very complementary. Would it be possible to add a figure with VIMS, ESO and ISS full-disk albedo at a phase angle of ~10° ?

Lines 215-216 : According to figure S6, there seems to be an inconsistency between ISS and VIMS in the CB3 channel as well. Can you comment on it ?

Lines 220-221 : I understand the reasoning for MT3 (the 3nm width of this ISS channel is too times narrower than the corresponding VIMS channel) but this reasoning does not stand for MT2 channel, for which the 15nm spectral resolution of ISS is coarser than VIMS.

Lines 251-252 and Fig S8: « ...which suggests the H-G function without specifying parameter range is better than the H-G function with specifying parameter range » : isn't it the opposite ?

Lines 274-275 and Fig S11 : similar comment than above : what about the inconsistency between ISS and VIMS in channels CB2 and CB3 ? Can you comment on that ?

Lines 305-306 : I do not understand how you interpolate the phase functions at the 12 ISS wavelength to all wavelength from 264 to 939nm : do you assume the spectral shape of the ESO albedo data (Karkoschka et al) ? Or the VIMS data ? What do you mean by the term « further combined » in the following sentence : « The interpolated phase function (0-180°) at each wavelength in the wavelength range of 264-939 nm is further combined with the available albedo with a phase angle 6.8° from Karkoschka » ?

Reviewers' comments:

Reviewer #1 (Remarks to the Author):

This paper reports on a re-analysis of the observational data available for the reflectivity of Jupiter at different wavelengths and phase angles. This allows for a calculation of Jupiter's albedo, and from that its internal heat source. The conclusion of this work is that this internal heat source is nearly 40% higher than previous estimates. This is a surprising new result that has important consequences for models of Jupiter's evolution, and by extension, for gaseous exoplanets in general. It is of great interest to the community and should be published.

Because of the importance of this result, I would like to see some of the steps in the process made more robust. For example, the SSI in the UV can be quite variable. Some statement should be made in the supplementary material as to the size of this variation and its contribution to the overall uncertainty in Jupiter's albedo and its internal heat source. In panel A of figure 17 in the supplementary material a comparison is made with the measurements of Karkoschka (1994, 1998). Although the agreement over most of the range is excellent, the difference between the 1993 (green) and 1995 (red) estimates are largest in

the visible wavelengths where the solar irradiation is the strongest. Are there other measurements that could be compared with as well? Is the temporal change indeed limited to these values or could it have been larger in the past? Can the data be compared with Hubble observations or any other spacecraft observations? I would urge the authors to look for at least one additional albedo measurement in the literature for a more thorough comparison. A small typo that caught my eye was in the supplementary material line 417 where instead of "production" it should say "product".

Reply: First, we correct the typo "production. Thanks for pointing out this!

Second, we think the reviewer provides very constructive suggestions. Yes, the possible temporal variation must be discussed with more details considering the large difference between our measurements in the Cassini epoch and the previous measurements in the Pioneer/Voyager epochs. By following the suggestions, we conduct more analyses and discussions of the temporal variation of Jupiter's albedo. In particular, we explore the long-term variations (less than a few percent), which is much smaller than the difference $\sim 47\%$ between the Cassini results and the Pioneer/Voyager results. We add the new discussions into a new Supplementary Information section (section 12 with a new figure Fig. S19). We also add more discussions in a new Supplementary Information section (section 11) about the reasons why the current Cassini measurements are more robust than the previous Pioneer/Voyager measurements by following the suggestions from you and from the other reviewer.

Reviewer #2 (Remarks to the Author):

In their paper, Li et al. provide updated estimates of Jupiter's internal heat, albedo and absorbed solar energy obtained from the analysis of (mainly) Cassini data acquired during the flyby of Jupiter in 2000-2001. These values are significantly different than that obtained from previous estimates (at the Pioneer and Voyager epoch) : Li et al derive a Bond albedo of 0.503 (compared to the previous estimate of 0.343) and consequently an internal heat flux of 7.485 W/m² (compared to 5.44 W/m²). These parameters are crucial to many studies: the history and evolution of the giant planets, their energy budget, their atmospheric circulation, etc.

Hence, the topic of this paper is relevant to be published in Nature Communications, as it has a broad impact. The results look convincing and the paper is rather well written, although the supplementary material is a bit long. Several concluding remarks about the internal structure of Jupiter or application to exoplanet atmospheres are a bit far-stretched or need more justifications. General and specific comments are detailed below.

General comments :

Several times in the paper and supplementary material, the Cassini measurements are referred to as having a « better » or « more complete » phase coverage compared to Pioneer measurements, but this point deserves to be better documented or quantified. Is it possible to give the reader an idea of the phase angles and wavelengths covered by

Pioneer measurements for instance ?

Reply: We add a new section in Supplementary Information (section 11) to discuss the difference of Jupiter's radiant energy budget between the new Cassini measurements and the previous Pioneer/Voyager measurements. In the new section, we provide more details about the differences between the Cassini observations and the Pioneer/Voyager observations (e.g., details of the coverage of phase angle and wavelength).

A related comment is about Figure 3 and its discussion lines 109-122 : is it possible to pinpoint the reason for the disagreement with the previously derived Bond albedo by Hanel et al ? For instance, is the geometric albedo in agreement with previous studies but not the phase integral, or the other way around ; or is this an effect of the wavelength coverage ?

Reply: Please see my reply to your above suggestion. In the new section (Supplementary Information 11), we provide more details of the differences between the Cassini observations and the Pioneer/Voyager observations. We also discuss why the new measurements are better than the previous measurements. Finally, we add one more section in Supplementary Information (section 12) by following the suggestions from the other reviewer, in which we provide more discussions on the temporal variations of Jupiter's radiant energy budget. With the discussions, we can rule out the possibility that the significant differences of Jupiter's radiant energy budget between the Cassini results and the Pioneer/Voyager results are due to the temporal variations.

Lines 164-168, regarding evolutionary and interior models : To my understanding, there are many arguments that suggest the existence of compositional gradients in giant planet interiors, leading to layered double diffusive convection (similar to the circulation in the Earth's oceans). To say that taking into account layered double diffusive convection is «potentially unnecessary » is a dangerous statement, as one cannot choose to discard a physical effect simply based on Li et al new results. Furthermore, lines 165-166, the statement that layered convection acts to « weaken the internal heat » is unclear. Indeed, layered double diffusive convection acts to slow the cooling of giant planet interiors and is used to explain, for instance, the excess of luminosity of Saturn (see for instance Leconte and Chabrier, Nature Geosciences 2013). Hence I do not understand how and why the revisited value of Jupiter's internal heat flux would make the layered convection theories obsolete.

Reply: Thanks for your suggestions and comments! Here is our understanding. The model of Leconte and Chabrier is entirely ad hoc. It merely posits a composition gradient throughout the interior of a Saturn, and investigates alternate evolutionary histories with such a gradient in place. It is really a thought experiment and a very extreme "non-standard" thermal evolution model of the planet. It has not been adopted as the standard within the community. While it is true that it has been suggested that composition gradients may exist near the core of the giant planets, based on formation models (Helled and Stevenson, 2017) or based on the miscibility of core materials (water, rock) in hydrogen (Militzer and collaborators) these are relatively recent suggestions and should be viewed as hypotheses. It is still extremely worthwhile to investigate whether a relatively "simpler" model -- more in line with standard practice over the past few decades and over dozens of

publications -- could explain the evolution of Jupiter and Saturn. Here we simply suggest that such a model, including He rain in both Jupiter and Saturn but with a simple structure aside from the rain, could possibly explain the thermal evolution of both planets, and it should be investigated in the future.

Furthermore, the particular setup advocated by Leconte and Chabrier in that paper has been shown to be in error by Vazan et al. (2016), who point out that such composition gradients are unstable to convection by the Ledoux criterion for triggering convection with composition gradients. In addition, the Leconte and Chabrier paper is silent on the issue of known He depletion in the atmospheres of both planets, so it cannot be viewed as an entirely viable and full alternative evolutionary history. That being said, a clear statement one SHOULD make is that the thermal evolution history of Jupiter should now be revisited, and that it is now far EASIER to understand the inferred He rain, since the intrinsic flux has now been increased by 40%. The problem has always been that previous has always showed that with He rain, Jupiter would be brighter than observed. This may no longer be the case!

One statement we want to emphasize is that the new results make it necessary to revisit some existing evolutionary models and theories of Jupiter, and we make this statement ONLY and remove the related discussions by referring your suggestions (please see lines 169-170 in the main text).

Lines 166-168 and in the abstract, the authors state that with their revised value for Jupiter's internal heat, « a consistent evolutionary history for both Jupiter and Saturn [...] can finally be obtained » : again, I find that this statement is a bit far-stretched without a model to support it. Secondly, just afterwards (lines 169-172) the authors advocate for the revision of the value of internal heat for Saturn (and other giant planets). These two statements are in contradiction :if Saturn's internal heat is revisited and a very different value is obtained, as suggested, then new models would be needed to explain the difference between Jupiter and Saturn heat budget and evolution...

Reply: A very good suggestion. We are still working on Saturn's radiant energy budget, and we agree that it is too early to say anything about Saturn before we finish the story of Saturn. So we follow this good suggestion to remove the statement "a consistent evolutionary history for both Jupiter and Saturn" from lines 166-168 and abstract (please see lines 39-40 and lines 169-170 in the main text).

I also do not understand the relevance of the comparison to cool giant exoplanets in the last paragraph. After all, Saturn's albedo has been estimated to 0.34, and we can expect a great diversity of albedo in extrasolar cool giant planets depending on their composition, cloud and aerosol properties ; or the stellar type. The fact that Cahoy et al., 2010 failed to match Jupiter's albedo is not relevant, as the authors modeled a « Jupiter-like » exoplanet and did not fine-tuned their atmospheric model to match Jupiter's cloud and haze properties. Indeed, Cahoy et al clearly indicate that they do not take into account NH₄SH cloud but only a NH₃ cloud (which might not be relevant) ; that their model lack haze layers ; and that their geometric albedo spectrum is clearly at odds and too bright compared to Karkoschka et al,

1994 albedo spectrum at wavelengths shorter than $0.5 \mu\text{m}$. Hence it is not surprising that they derive a bright geometric albedo for their « Jupiter-like » planet. For these reasons, I recommend removing the reference to Cahoy et al., 2010.

Reply: We follow the suggestions by removing the reference and the corresponding discussion (please see lines 186-189 in the main text).

Specific comments in Supplementary material :

Supp. Info 3 : A reference to Figure 1 is missing

Reply: We correct it (please see line 86 in Supplementary Information).

Line 172 : « The Cassini spacecraft has one more instrument » : one more camera (or imaging system) instead ?

Reply: We follow it (please see line 173 in Supplementary Information).

Figure S5 is very interesting but it is hard to see how well the VIMS and ESO data compare at a similar phase angle (about 10°). Actually, a comparison with the 12 channels of ISS at the same phase angle would be a great addition. Indeed, the following figures compare ISS and VIMS albedo versus phase angle at different wavelength, but the comparisons of the albedo spectra at a given phase angle would be very complementary. Would it be possible to add a figure with VIMS, ESO and ISS full-disk albedo at a phase angle of $\sim 10^\circ$?

Reply: By following this very good suggestion, we add a new figure (Fig. S7) in Supplementary Information (section 6). In addition, we add the corresponding discussions in lines 212-255 in this section.

Lines 215-216 : According to figure S6, there seems to be an inconsistency between ISS and VIMS in the CB3 channel as well. Can you comment on it ?

Reply: We add the discussion on the difference with the new figure (Fig. S7) in Supplementary Information (lines 212-255). The calibration of the Cassini ISS and CIRS is not perfect, and the Cassini teams are still working on it (we mention this point in lines 236-241 in Supplementary Information section 6).

Lines 220-221 : I understand the reasoning for MT3 (the 3nm width of this ISS channel is too times narrower than the corresponding VIMS channel) but this reasoning does not stand for MT2 channel, for which the 15nm spectral resolution of ISS is coarser than VIMS.

Reply: We do not have good explanation for the difference at MT2 and other filters. As we reply to your above comments, the Cassini ISS/VIMS teams are still working on the calibration issue (please see lines 236-241 in Supplementary Information).

Lines 251-252 and Fig S8: « ...which suggests the H-G function without specifying parameter range is better than the H-G function with specifying parameter range » : isn't it the opposite ?

Reply: Many thanks for pointing the mistakes. Yes, there are mistakes in the legend for the blue and green lines in Fig. S8 (now Fig. S9). The green and blue lines are H-G fitting WITH

and WITHOUT specifying parameter range, respectively. We correct the mistakes in the new figure.

Lines 274-275 and Fig S11 : similar comment than above : what about the inconsistency between ISS and VIMS in channels CB2 and CB3 ? Can you comment on that ?

Reply: As we reply to your previous comments, the calibration of the Cassini ISS and VIMS is not perfect, and the Cassini teams are still working on it (see lines 236-241 in Supplementary Information).

Lines 305-306 : I do not understand how you interpolate the phase functions at the 12 ISS wavelength to all wavelength from 264 to 939nm : do you assume the spectral shape of the ESO albedo data (Karkoschka et al) ? Or the VIMS data ? What do you mean by the term « further combined » in the following sentence : « The interpolated phase function (0-180°) at each wavelength in the wavelength range of 264-939 nm is further combined with the available albedo with a phase angle 6.8° from Karkoschka » ?

Reply: Yes, we assume the spectral shape of the ESO albedo data. We make it clear in the revision (please see lines 330-336 in Supplementary Information 8).

REVIEWERS' COMMENTS:

Reviewer #2 (Remarks to the Author):

I have carefully read the new version of L. Li's manuscript " Less Absorbed Solar Energy And More Internal Heat For Jupiter" along with the replies to the referee's comments and find that this new version addresses well the comments made by the referees.

I only found one typo line 493 of the supplementary material: "emptied power" should be "emitted power".

Hence, I recommend its publication without any further modifications.

Reviewer #2 (Remarks to the Author):

I have carefully read the new version of L. Li's manuscript " Less Absorbed Solar Energy And More Internal Heat For Jupiter" along with the replies to the referee's comments and find that this new version addresses well the comments made by the referees.

I only found one typo line 493 of the supplementary material: "emptied power" should be "emitted power".

Hence, I recommend its publication without any further modifications.

Reply: We corrected the typo "emptied".